# Adoption of Large-Scale Scrum Practices through the Use of Management 3.0

Fernando Almeida [1,*] and Eduardo Espinheira [2]

1 Polytechnic Higher Institute of Gaya (ISPGAYA) and INESC TEC, 4400-103 Porto, Portugal
2 QAValue and Porto Business School (PBS), University of Porto, 4460-314 Porto, Portugal; eespinheira@pbs.up.pt
* Correspondence: almd@fe.up.pt

**Abstract:** Software engineering companies have progressively incorporated agile project management methodologies. Initially, this migration occurred mostly in the context of startups, but in recent years it has also sparked interest from other companies with larger and more geographically dispersed teams. One of the frameworks used for large-scale agile implementation is the LeSS framework. This study seeks to explore how Management 3.0 principles can be applied in the context of the ten practices proposed in the LeSS framework. To this end, a qualitative research methodology based on four case studies is used to identify and explore the role of Management 3.0 in software management and development processes that adopt this agile paradigm. The findings show that the principles of Management 3.0 are relevant to the implementation of the LeSS framework practices, especially in fostering team values and personal values; however, distinct principles between the two paradigms are also identified, namely the greater rigidity of processes advocated in the LeSS framework and a greater focus on process automation.

**Keywords:** software engineering; agile; Scrum; project management; self-managed teams; leadership





## 1. Introduction

Agile management has transformed the way projects are planned, executed, and monitored, which has caused many companies to start using it. As Sirashki [1] notes, agile is not only related to speed, but also to flexibility and integration. As such, agile project management methods are adaptable to sudden changes in planning, which allows for changing priorities, postponing tasks, and changing project features as needed.

Agile methodologies advocate a set of values that promote organizational models based on people, collaboration, and communities working for motivation. People are precious resources not because they are seen only as innate resources, but because they are carriers of values and culture [2]. Hohl et al. [3] add that these methodologies require that developers, customers, and managers alike change the way they work and think. These changes must occur naturally in the context of each organization and do not come about by imposition.

Management methods keep up with current social and technological trends. These changes are currently visible, as Khairullah [4] notes that companies with a horizontal culture and collaborative management attract more attention from a new generation entering the job market. Guided by the opportunity for growth and transparency in relationships, collaborative management presents itself in a democratic and inclusive way, captivating different talent profiles and stimulating innovation as recognized in [5].

Management 3.0 emerges as one of the ways to implement collaborative management. This paradigm is a management methodology that decentralizes decision making. Unlike the traditional model, where there is a leader responsible for dictating the rules and strategies alone, in collaborative management, everyone contributes to reaching decisions [6]. In

Management 3.0, the focus is on people who are the greatest asset of an organization [7]. While in vertical management, the leader concentrates the decision power and responsibility for an entire team; in the horizontal model, all employees have a voice to contribute to long-term planning. Furthermore, Management 3.0 aims to engage employees, and thus leverage business growth. All employees work towards a common goal of fast, agile and effective delivery [8].

Management 3.0 emerged only in 2011, and it is a paradigm that has been applied mostly in the context of young and small companies that adopt agile management processes. Empirical studies have mostly highlighted its relevance in the context of software companies [9–11], but other studies have also shown the success of this management model in other business sectors [12,13]. Furthermore, agile methodologies are especially suited to work in small, cross-functional, and collaborative teams [14]; however, as Sablis et al. [15] highlight, many projects have high complexity and require the involvement of a large number of collaborators, many of whom may be geographically distributed. The adoption of Management 3.0 in larger software companies that have large-scale agile teams is a topic that has not been explored in the literature. In this sense, exploratory studies are needed to explore the relevance of Management 3.0 practices in the context of large-scale agile teams that adopt the Large-Scale Scrum (LeSS) framework. It is intended that this study can contribute to Management 3.0 being effectively used in the context of large-scale teams by reducing the risk of misalignment and low team motivation and performance, as recognized in Conboy & Carroll [16] as the main factors inhibiting the application of LeSS.

The rest of this manuscript is organized as follows: Initially, a theoretical contextualization on the migration processes from agile to large-scale and on the practices implemented in the LeSS framework is performed. Next, the methodology adopted in this study is presented, and a presentation of the case studies participating in this study is also performed. After that, the results of the study are provided considering the ten specific practices incorporated in the LeSS framework, and the relevance of Management 3.0 in the implementation of the LeSS framework is also discussed. Finally, the conclusions are summarized and are followed by a brief presentation of the main limitations of the study and indications for future work.

## 2. Background

### 2.1. From Small-Scale to Large-Scale Agile

The agile method serves as a common label for many well-defined procedures that call themselves agile and vary in practice. One way to explore agile methods is to look at the main characteristics and practices that they share. Dingsøyr et al. [17] state that agile methods are very lightweight processes that employ short iteration cycles. Furthermore, they seek to actively involve users to establish, prioritize, and verify requirements [17]. In this sense, agile methods are a consequence of the experience of prototyping and rapid development. This clarification of the concept emerged only in 2001 through the Agile Manifesto, in which four fundamental principles are advocated: valuing individuals and interactions as opposed to processes and tools, functional software over comprehensive and exhaustive documents, collaboration with the customer over contractual negotiation, and responding to changes rather than following a plan [18].

Developing software is a highly complex activity and the means that are used to build the final product are extremely volatile. Poorly formulated and dynamic user requirements, the high number of people involved in the process, and the interaction of the program being developed with others in the same system contribute to this complexity [19,20]. San Cristóbal et al. [21] add that complex problems are usually difficult to predict or are even unpredictable. In software, we encounter this situation, and in an attempt to circumvent this unpredictability, an adaptive process such as incremental and iterative development, is used. The ability to deal with the unforeseen events typically found in the software industry, associated with the lightness of the process, focus on periodic delivery of a system, and constant improvement makes agile models very attractive to both companies and

their customers [22]. At this level, it also becomes relevant to consider that for a long time, software development was supported by the traditional waterfall model; however, its inability to manage complexity and unforeseen events led to the emergence of new processes supported by the agile software development paradigm.

Large companies have been forced to change the way they develop software, leaving traditional methods and embracing agile methods to increase productivity and software quality [23,24]; however, implementing agile methods in large enterprises is not always easy and involves embracing a new organizational culture, which can bring major obstacles [25]. Furthermore, managing tasks and multiple teams becomes difficult for organizations, which need to coordinate the work of these teams, which in many cases are geographically distributed [26]. To minimize this impact and disseminate these new methods, large organizations usually create small teams and gradually disseminate this new way of working. Starting from this point, it becomes relevant to look at how to scale agile to the entire organization.

Agile methods have transformed the way software has been developed by emphasizing user involvement, small teams, high agility, and tolerance for change [27–29]. This flexibility has contributed to large organizations applying agile methods to projects of greater complexity, aiming to achieve high levels of performance; however, the best-known agile methods such as Scrum, Lean Software Development, or XP were initially designed for small and centralized teams.

In contrast, large-scale projects of greater complexity require a greater volume of resources (e.g., team size, code, budget, or employees). This creates an added difficulty in achieving satisfactory levels of quality and performance. Recent literature has addressed these challenges. In Uludag et al. [30], difficulties arising from the interconnection between multiple programs is recognized; in Dingsøyr et al. [31], the difficulty of coordinating work involving multiple teams working on a single product is highlighted; while in Shameem et al. [32], the added difficulty of having teams that are distributed across multiple geographic locations is further mentioned. In conclusion, scaling agile practices is not an easy task in a short period of time.

### 2.2. The LeSS Framework

The LeSS framework was proposed in 2016 by Craig Larman and Bas Vodde, and sought to help large organizations use Scrum on a large scale, namely considering geographically distributed teams [33]. A key pillar of LeSS is to apply the principles of Scrum at a large scale. Similar to Scrum, LeSS is also a framework designed to deal with incomplete scenarios, and uses the empirical approach of transparency, inspection, and adaptation to work, rather than seeking the illusory predictability that allows predicting the functioning of organizations. In this sense, and as pointed out in [33], LeSS seeks to simplify a large and complex organization. Large-scale Scrum can be implemented according to the LeSS framework, which scales Scrum up to eight teams (e.g., eight people maximum in each), and Less Huge, which allows scaling up to a few hundred or thousand people in a single product.

In Alsaqaf et al. [34], it is highlighted that the search for simplicity must be accompanied by a change in mindset that must move from a project-focused mindset to a product-focused mindset. In this sense, the organization moves the focus of its work from the continuous delivery of projects, which often represent large batches of work, to the incremental delivery of value to its customers. The product becomes continuously evolving sprint after sprint. Furthermore, LeSS eliminates the need for portfolio management by considering a single Product Backlog [35]. The work, performed in small batches, is continuously integrated between the various teams, allowing for frequent deliveries to the users of the product. These short cycles continuously provide feedback for planning and maximizing the organization's adaptability. Furthermore, when implementing LeSS, many cross-functional teams are recommended.

LeSS gives the implementer a lot of freedom by not establishing a set of rigid rules. Only guidelines for organizational structure, product management, and working with multiple teams in a single sprint are established. This freedom regarding rules is only possible due to the principles established by the framework that help organizations in its implementation. As Auerbach & McCarthy [36] point out, the Lean approach is present in the implementation of agile; therefore, the framework seeks the removal of waste from the production process over a search for productivity improvements in resource utilization. The empirical control of the process and its continuous improvement causes the product to be developed in short cycles, which allows it to be adapted and corrected in each cycle [37]. The focus of LeSS is on the scalability of Scrum, but it is argued that this goal is compatible with maintaining customer focus throughout the organization [38].

LeSS presents a set of practices that are interconnected and depend on each other and amplify each other. LeSS follows an evolutionary approach, rather than advocating one large architecture and initial design. Implementing this approach requires that teams are comfortable with altering code and changing architecture without affecting its operation. Below is a summary of the ten practices proposed by LeSS [33]:

- Acceptance Tests (AT)—acceptance testing should represent the users' point of view. This approach gives developers a direct insight into what customers want and how the product will be used; thus, it is possible to avoid ambiguity in the process and reduce the chances of major mistakes being committed;
- Architecture & Design (AD)—it supports the paradigm that design and architecture are separate components and also advocates the creation of growing and evolving design;
- Clean Code (CC)—development of functional code is not a sufficient condition. The code must have quality from planning to execution, which will facilitate and reduce the time associated with code maintenance;
- Continuous Delivery (CD)—deliveries are made in a predictable, frequent, and automated way. This approach provides greater control over product quality;
- Continuous Integration (CI)—code integration is performed as often as features are developed. The main goal is to quickly check that changes or new features have not created new defects in the project;
- Specification by Example (SE)—a set of practices that help build a product in the right way, focusing on communication between all parties involved, ensuring that everyone has a clear understanding of what is being produced and can collaborate as effectively as possible;
- Test-driven Development (TDD)—it is advocated that the test should be written before the code. This approach makes it possible to quickly identify errors in the code and to fix them quickly;
- Test-driven Thinking (TDT)—it is advocated that testing should be incremental and interactive, where each developed feature is considered ready only after due testing has taken place;
- Tests Automation (TA)—one way to make software testing more independent compared to human intervention is to use automated testing as a best practice. Automated testing can capture behavior and feedback in an automatic and dynamic way;
- Unit Tests (UT)—the aim is to verify the behavior of the smallest units of the product in a fast and automated way. These tests need to run in isolation because they need to run fast and as soon as possible.

## 3. Materials and Methods

Scaling agile in companies requires different actions, since, without a clear structure, the process of scaling this methodology presents significant risks and a low probability of success, as recognized in the studies conducted by Kalenda et al. [39] and Robert [40]. Furthermore, in Annosi et al. [41], it is also recognized that the large-scale agile approach can be an inhibitor of innovation and learning capacity if these practices are too rigid. The

lessons learned from these studies allow us to conclude that there is no single correct way to apply agile at scale, because each company has its own particularities, and therefore, they sometimes develop their own processes, teams, and cultures. In this sense, and to capture the specificities and unique characteristics of each organization, a qualitative methodology based on case studies was considered. This approach allows us to explore contemporary events in their natural context; however, as recognized in Ebneyamini & Moghadam [42], the results strongly depend on the integration capacity of the researcher.

In this study, and in order to reduce the risk of bias, multiple case studies were considered. In all, four case studies of software companies located in Portugal were included, although they develop solutions for both national and international markets. More interesting than the similarities that could be found by including multiple case studies, their greatest potential lies in the possibility of finding differences and exploring their explanations. In Yin [43], two approaches to implementing multiple case studies are acknowledged, namely the literal replication model (e.g., where the chosen cases are expected to have similar results), and the theoretical replication (e.g., where the chosen cases are expected to have contrasting results, for predetermined reasons); thus, in this study, we chose to adopt contrasting case studies involving two case studies of small and medium enterprises (SMEs), which have less than 250 employees, and two large enterprises (LEs), which have more than 250 employees. The profiles of the companies that participated in this study are presented in Table 1. All companies implement the LeSS framework and promote the inclusion of Management 3.0. Furthermore, they operate in the software development area, although there are differences between them, namely in terms of size, area of operation, and business models. This diversity of characteristics among the organizations participating in this study allows us to explore the relevance of Management 3.0 in the context of each of the practices included in the LeSS framework.

**Table 1.** Presentation of the case studies.

| Case Study | Founding Year | Size | Description |
|---|---|---|---|
| CS1 | 1998 | SME | Software development company that operates mainly in the Portuguese-speaking markets (e.g., Angola and Brazil). The company focuses mainly on developing software solutions for the public sector. Solutions are developed in the areas of public procurement, accounting, and document and process management. The implementation of the LeSS framework arose out of the need to involve geographically distributed Scrum teams larger than 15 and 20 members. Initially, the company worked in a waterfall environment, and the migration to the LeSS framework occurred without previous experience in Scrum. |
| CS2 | 2014 | SME | The company has 8 years of activity and started its activity in the web design field. Later, and with structural changes in this area, the company expanded its activity to cover other areas such as the marketing of computer equipment, software marketing, automation development. Currently, most of the projects developed by the company are in the field of IoT integration in home automation solutions and their incorporation with mobile devices. Since its conception, the company implemented the Scrum methodology. However, this framework proved to be insufficient, given the growth of the company and the involvement of employees from multidisciplinary areas. The migration to the LeSS framework occurred only in early 2020, in the context of the COVID-19 pandemic. |

**Table 1.** *Cont.*

| Case Study | Founding Year | Size | Description |
|---|---|---|---|
| CS3 | 2003 | LE | The company operates in the global market through e-marketplace solutions and has software development teams in Portugal, Spain, and Brazil. Initially, the company started by developing virtual stores for small retailers that needed to have an online presence and helped them in the digital transformation process. With the acquisition of new skills, the business model migrated to e-marketplaces, in which several different vendors or companies offer their products or services on the platform. The company started by adopting Scrum in small local teams in each of the countries with pilot projects. The success of these initiatives led to the model being replicated across multiple teams. LeSS emerged from the need to integrate the work of these teams. |
| CS4 | 2009 | LE | A company that has adopted the software as a service (SaaS) model since its inception. The company operates in the global market, providing services mainly to the Asian market. In implementing its solutions, the company offers a business model tailored to each client. The company takes responsibility for security, maintenance, and system updates, making the solution even more complete. LeSS arose from the need to integrate the work of several teams, some of which were operating in an outsourcing model. The need to have greater visibility on the work of these teams was the fundamental trigger for joining LeSS in 2017. |

The interviews were conducted between June and December 2021 through an online meeting platform (e.g., Zoom, Google Meet, and MS Teams) with the Chief Technology Officer (CTO) of each company. The adoption of digital means to conduct these interviews was indispensable in view of the containment and restriction measures caused by COVID-19. Each interview lasted between 30 and 45 min. The interviews were later transcribed and validated by the interviewees. The interviewees were asked about the dynamics and benefits brought by Management 3.0 for each of the LeSS framework practices as presented in Table 1. Accordingly, ten questions were asked in the interviews that correspond to the total number of LeSS framework practices.

The analysis of the interviews was performed using webQDA, which is a qualitative data analysis and exploration software. webQDA uses thematic analysis to find matching themes in the responses. Thematic analysis seeks to identify patterns in the data by implementing the technique of induction or deduction [44]. Inductive analysis is guided by the data, without attempting to fit into a pre-existing coding model or analytical biases of the researcher; however, since a theoretical or deductive analysis is consequently more explicitly analyst-driven, this form of analysis tends to describe the data less in general, highlighting in more detail only certain aspects of the data. This study adopted a deductive analysis in which the pattern categories were loaded into webQDA, as presented in Figure 1. A total of 72 themes were considered, as proposed by Appelo [7], of which six of these themes correspond to Management 3.0 principles (i.e., align constraints, empower teams, energize people, develop competence, grow structure, and improve everything). The description of the methodological process is presented in Figure 2. Initially, the 72 previously mentioned themes are loaded into webQDA. After that, the software is responsible for finding occurrences in the four case studies. Finally, the number of occurrences is counted, and the themes are sorted in descending order of their incidence level. The degree of homogeneity of each theme found is also analyzed, and, for this purpose, its standard deviation is calculated. A high value of the standard deviation indicates high levels of asymmetry among the case studies, while a low value indicates that the theme is typically found homogeneously across all case studies.

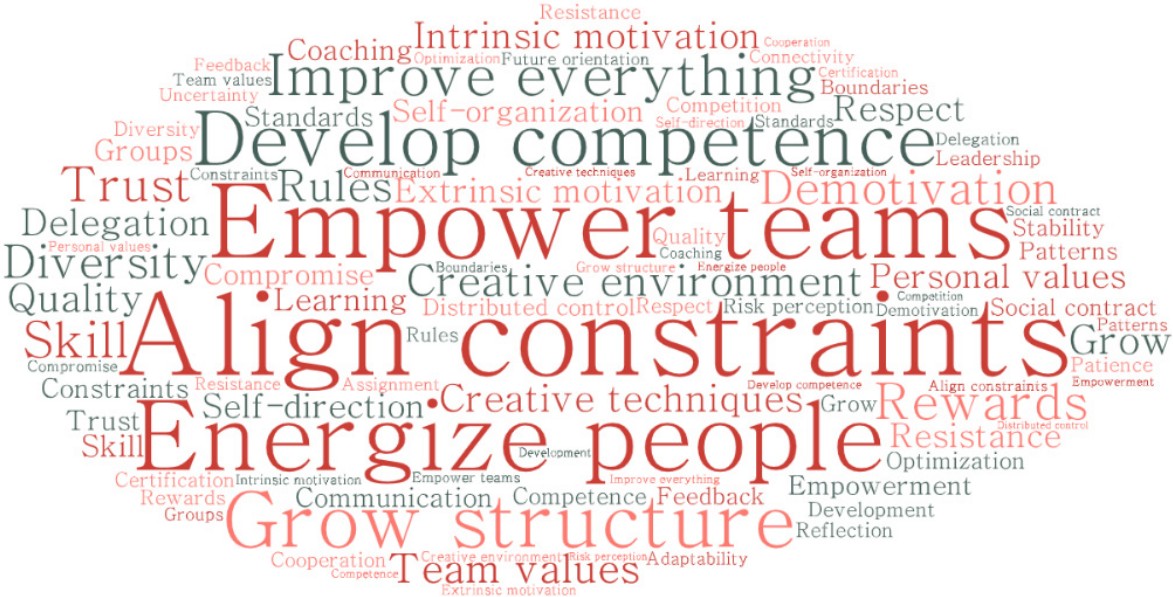

**Figure 1.** Cloud of themes associated with Management 3.0.

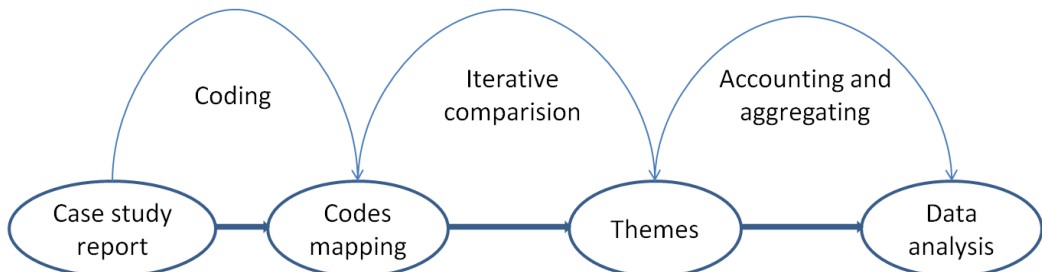

**Figure 2.** Adoption of the thematic analysis methodological process.

## 4. Results

Table 2 presents the final themes identified in webQDA by each LeSS practice. The table shows the percentage of themes found by each practice considering the individual view of each case study, the total number of occurrences (NO), and the standard deviation (SD) of their distribution. The information regarding the NO is important to understand the level of incidence of the principles proposed by Management 3.0, and the SD allows us to understand their degree of homogeneity. NF means that the theme was not found. It is important to note that the same theme may occur more than once for each case study, and, in such occurrences, it was mentioned several times in the answer provided by the respondent. Additionally, some themes appear in more than one LeSS practice. Unmatched themes are not represented.

**Table 2.** Correspondence of final themes.

| Theme | CS1 | CS2 | CS3 | CS4 | NO | SD |
|---|---|---|---|---|---|---|
| *AT* | | | | | | |
| Assignment | 0.2857 | 0.1429 | 0.2857 | 0.2857 | 7 | 0.5000 |
| Feedback | 0.2000 | 0.2000 | 0.2000 | 0.4000 | 5 | 0.5000 |
| *AD* | | | | | | |
| Connectivity | 0.1667 | 0.3333 | 0.1667 | 0.3333 | 6 | 0.5774 |
| Standards | 0.2000 | 0.2000 | 0.2000 | 0.4000 | 5 | 0.5000 |
| Patterns | 0.2000 | 0.2000 | 0.2000 | 0.4000 | 5 | 0.5000 |
| Boundaries | NF | NF | 0.5000 | 0.5000 | 4 | 1.1547 |
| Risk perception | NF | NF | 0.3333 | 0.6667 | 3 | 0.9574 |
| *CC* | | | | | | |
| Team values | 0.3333 | 0.1667 | 0.1667 | 0.3333 | 6 | 0.5774 |
| Personal values | 0.1667 | 0.1667 | 0.1667 | 0.5000 | 6 | 1.0000 |
| *CD* | | | | | | |
| Team values | 0.3333 | 0.1667 | 0.1667 | 0.3333 | 6 | 0.5774 |
| Personal values | 0.1667 | 0.1667 | 0.3333 | 0.3333 | 6 | 0.5774 |
| Quality | 0.2500 | 0.2500 | 0.2500 | 0.2500 | 4 | 0.0000 |
| Feedback | 0.2500 | 0.2500 | 0.2500 | 0.2500 | 4 | 0.0000 |
| *CI* | | | | | | |
| Team values | 0.3333 | 0.1667 | 0.1667 | 0.3333 | 6 | 0.5774 |
| Personal values | 0.1667 | 0.1667 | 0.3333 | 0.3333 | 6 | 0.5774 |
| Quality | 0.2500 | 0.2500 | 0.2500 | 0.2500 | 4 | 0.0000 |
| Grow | NF | NF | NF | 1.0000 | 1 | 0.5000 |
| *SE* | | | | | | |
| Cooperation | 0.2000 | 0.2000 | 0.4000 | 0.2000 | 5 | 0.5000 |
| Empowerment | 0.2500 | NF | 0.2500 | 0.5000 | 4 | 0.8165 |
| Self-organization | NF | NF | 0.5000 | 0.5000 | 2 | 0.5774 |
| Reflection | NF | NF | NF | 1.0000 | 2 | 1.0000 |
| *TDD* | | | | | | |
| Development | 0.2222 | 0.1111 | 0.2222 | 0.4444 | 9 | 1.2583 |
| Adaptability | NF | NF | 1.000 | NF | 3 | 1.5000 |
| *TDT* | | | | | | |
| Interactivity | 0.2000 | 0.2000 | 0.2000 | 0.4000 | 5 | 0.5000 |
| Reflection | NF | NF | NF | 1.000 | 1 | 0.5000 |
| *TA* | | | | | | |
| Optimization | 0.1667 | 0.1667 | 0.2500 | 0.4167 | 12 | 1.4142 |
| Stability | 0.2857 | NF | 0.2857 | 0.4286 | 7 | 1.2583 |
| Rules | 0.6667 | NF | 0.3333 | NF | 3 | 0.9574 |
| *UT* | | | | | | |
| Assignment | 0.1667 | 0.1667 | 0.3333 | 0.3333 | 6 | 0.5774 |
| Optimization | 0.2500 | NF | 0.2500 | 0.5000 | 4 | 0.8165 |
| Patience | NF | 1.0000 | NF | NF | 1 | 0.5000 |

The findings reveal the emergence of different themes for each LeSS practice, respectively:

- AT: Management 3.0 practices are relevant in assigning tests to team members and in-process feedback. Both have identical relevance for the four case studies;
- AD: Connectivity, standards, and patterns are fundamental elements advocated in Management 3.0 and are relevant in this LeSS practice. Equally relevant, and which stands out in the context of large companies, are the system boundaries and the perception of risk. This situation explains a greater SD for these two Management 3.0 principles;
- CC: The alignment of team values and personal values is equally important to have a clean code; nevertheless, personal values are more relevant for CS4;
- CD: In this dimension, both team values and personal values stand out; also relevant is the relevance of this practice for software quality and feedback for all companies;

- CI: All the themes identified in the "CD" practice stand out; however, "grow" theme was identified in CS4 as an element that promotes the emergence of potentially more complete solutions;
- SE: Cooperation is a fundamental element for teams in LeSS. Empowerment is another element that stands out and in which it is sought that the teams manage to implement the solutions with sufficient autonomy. With less weight comes the self-organization of teams (CS3) and reflection on the implemented processes (CS4);
- TDD: Test-driven development is the main element of this practice. Adaptability is a feature that was highlighted only in CS3;
- TDT: Interactivity is the fundamental element identified in all case studies, followed by reflection, which was only identified in CS4 with a single occurrence;
- TA: Optimization is a core element in the process of implementing this practice in LeSS and advocated in the context of Management 3.0. This is the theme with the highest number of occurrences (NO = 12), despite its themes having a high SD. The contribution of this practice to stability is also important, although it was not identified in CS2. More relevance is given to the role of rules;
- UT: The assignment of this task to team members was identified in all case studies. The contribution of these tests to optimization is also widely recognized, except for CS2, in which patience stands out (despite having only one occurrence). Patience is understood as the ability of employees to implement a repetitive process.

Finally, there are themes that are relevant in the context of Management 3.0 but do not come up as associated with any LeSS practice, such as the existence of a creative environment in which creative techniques can be adopted or the role of motivation, both extrinsic and intrinsic.

## 5. Discussion

The findings reveal that the guiding principles of Management 3.0 are relevant in the adoption of the LeSS framework. Management 3.0 principles are identified in all practices of this framework. As Appelo [7] highlights, Management 3.0 aims to change the way the leader or management acts, and for this to happen, the leader needs to learn new methods and practices for adopting agility to face business challenges and make the team generate better results. Kincius [45] adds that in implementing Management 3.0, the manager needs to keep in mind that people and their team should not be managed, but one should above all promote the environment in which the various players operate. This means that the leader must provide all the conditions for the activities to be carried out in the best possible way by the employees. The implementation of Management 3.0 contributed to reducing the resistance to change caused by the introduction of a framework such as LeSS, as recognized in Dikert et al. [46] as one of the main inhibitors of the transition to Scrum on a large scale in organizations. In CS3, this model is evident by highlighting that consulting processes are customized according to the clients' digital transformation challenges. Here it is fundamental that consultants have the freedom to understand the context of each organization and assess its level of digital maturity, so that the solutions adopted can meet the needs and installed skills of their clients.

The LeSS framework inherits the same characteristics as basic Scrum and advocates that each team should be small (e.g., between three and nine members). It also advocates that teams should be self-organized and multidisciplinary [47]. This is evident in CS2 in which small teams are used (e.g., ideally between 4 and 5 employees) and in which multidisciplinary profiles in the area of software engineering, web design, and home automation are integrated. In CS2, the empowerment of teams in the construction of solutions is also promoted. In this way, the top-down and bottom-up approaches of the LeSS implementation processes as advocated in Conboy & Carroll [16] are mixed, and thereby a greater employee adherence to the framework is achieved. This is also precisely a point where there is a total alignment between the principles advocated in the LeSS framework and Management 3.0, in which the empowerment of teams is promoted. With this, it is

intended to transform the workplace into a transparent and collaborative environment, giving conditions for each one to make their own decisions and take responsibility for them. The performance must be multidisciplinary, in which the individual must not only do what he or she has been proposed to do, or only what he or she proposes to do, creating an environment in which the collaborator can perform in the best way possible and be happy with it; however, the self-organization model may not work. To reduce this risk, Srivastava & Jain [48] advocates that a clear purpose and shared goals are important. In Silva et al. [49], it is further highlighted that learning from mistakes and always seeking to improve should be a rule. The structure should grow, but consciously, without causing damage to the quality of the corporate environment. It is about optimizing processes and motivating workers by making everything more productive. This is a vision also present in the principles of the LeSS framework through the "system thinking" approach, which advocates that one should understand and optimize the system as a whole, and not as a sum of different individual parts; however, the risk of lower specialization of employees in a given area may reduce their knowledge accumulation and integration, as recognized in [41]. Management 3.0 may contribute to mitigating this risk by offering greater autonomy to teams and reducing rigid and fixed meetings that may lead to lower levels of team productivity.

Management 3.0 states that organizational values should not be predetermined and independent of the team's structure; therefore, the leader must take on the challenge of getting to know their team members. In Appelo [7], several practices are suggested, such as Kudo Cards (e.g., cards that promote greater integration of team members through thanks or praise), Delegation Poker (e.g., cards to assist in visualizing the responsibilities assigned to each member), or Moving Motivators (e.g., cards to promote team synergy through greater knowledge about the motivating elements of its members). Another important point to note is that motivators are very personal, and this should be taken into consideration when energizing people, understanding how each one activates their motivators, and designing strategies according to the team's individual and collective profile. Furthermore, it is advocated that teams choose their values according to the current context and personalities, which needs to be revisited with some frequency, since we are in constant transition. Finally, the leaders' values should match those of the team so that leadership is by example. As acknowledged by Crevani et al. [50], the current digitalization challenges caused by COVID-19 and the high competitiveness of markets on a global scale make leadership processes even more relevant. To lead is to know how to achieve results through people, even in contexts of change. This implies valuing motivation and the creation of a favorable relational and work environment. CS4 recognizes its importance, whether in professional or even personal life. The ability to lead is always being tested, as a leader is a leader in the most diverse environments of his or her life. He also adds that not adopting an inspiring and trustworthy stance in any environment means personality weaknesses, which is in line with the studies developed by Jaroliya & Gyanchandani [50] and Khan et al. [51] in the organizational leadership field. In this sense, leading by example requires the leader to have a reliable and inspiring attitude, because their followers will analyze every detail of the leader's attitude and personality to draw the necessary inspiration to continue in the achievement of positive results.

Although the findings show a great relevance of the principles advocated in Management 3.0 in the implementation of the ten practices of the LeSS framework, there are also some differences and less alignment in some areas. The LeSS framework has been mostly implemented in large software companies with geographically distributed teams, while Management 3.0 is more easily found in young companies such as startups with a small number of employees. This causes the LeSS framework to have to include some rigidity in the standardization of processes. This also occurs in Management 3.0 in the implementation of the principle concerning the alignment of constraints. As is highlighted in Lawrie et al. [52], for a complex organization to function within a networked management model, everyone must be aligned with organizational goals; however, the concept

of aligning constraints is not linked to restricting the powers of the team. Moreover, the LeSS framework establishes concrete practices in implementing Scrum on a large scale by highlighting the role of testing in this process and of continuous integration and delivery. There is a clear emphasis on process automation [53–55]. Automation of processes in Management 3.0 takes a back seat, with priority given to the freedom of employees to experiment properly framed in an environment guided by transparency and collaboration that is conducive to employees taking responsibility for their decisions, as well as developing a multidisciplinary performance profile.

## 6. Conclusions

This study reveals that Management 3.0 is not a framework, but a set of principles for managing teams and organizations, and consequently, there is no single and complete way for its practical implementation in the short term. Since it is a way of reflecting on management, it is up to leaders to exercise the listed principles daily; therefore, one of the most important measures to make it real in companies is to create a scenario in which everyone involved feels comfortable to think, decide, and innovate.

Despite the freedom of implementation advocated in Management 3.0, in all case studies covered in this study, it was possible to identify that several Management 3.0 principles are relevant in the implementation of LeSS framework practices. These principles present in Management 3.0 prove to be adequate in the implementation of the LeSS framework in the context of an SME or LE. The team values and personal values are fundamental in the implementation of clean code, continuous delivery, and continuous integration.

This study also reveals that although there are clear synergies between Management 3.0 and the LeSS framework that allow LeSS implementation to be enhanced through greater team integration and autonomy, there are also important differences that it is important to recognize. LeSS establishes a greater rigidity of processes to allow several teams to work on the same product. This product brings end-to-end solutions with a focus on its end-users, that is, those who will use it, rather than on components, layers, or intermediate steps. However, this approach does not preclude the different teams from collaborating with each other as needed, especially to resolve any dependencies that are identified. Another difference arises in the automation of processes, which is a central element in the implementation of LeSS, and has the side effect of reducing impediments, whereas, in Management 3.0, it is argued that these impediments can be more easily mitigated through the alignment of constraints, in which it is argued that when each individual feels empowered and energized, they will tend to focus efforts on meeting organizational demands; therefore, the process of aligning constraints is the vision by which a manager will be able to deal with even divergent interests in favor of one cause only.

This manuscript offers both theoretical and practical contributions. In the theoretical dimension, it should be mentioned that the topic of agile implementation on a large scale has been little explored since the main studies have mainly focused on the implementation of agile in small teams that share the same organizational environment. Furthermore, studies concerning the implementation of LeSS have mostly looked at the processes of replicating Scrum from a single team to multiple teams (e.g., at local and distributed levels). Moreover, the main focus has been the comparative analysis of the LeSS framework with other alternative models such as SAFe or Nexus are noteworthy. This study takes a different perspective by looking at how practically LeSS can be enhanced by the adoption of Management 3.0. In the practical dimension, the results of this study are mainly aimed at two types of companies: (i) organizations with high levels of maturity in adopting agile on a large scale but are unaware of the potential of the principles advocated by Management 3.0; (ii) smaller organizations that have been committed to implementing Scrum and Management 3.0 since its inception but have recently experienced significant growth and, consequently, they need to migrate to a full-scale agile framework to reduce the migration risks and integrate the work of several times.

Finally, it is important to look at some limitations of this work. It is noteworthy that the qualitative approach through the realization of four case studies allows us to know in greater depth the way the LeSS framework is implemented in these organizations; however, it has obvious limitations in the generalization of the results. In this sense, and for future work, it would be interesting to consider a study methodology based on mixed methods, in which the implementation of the LeSS framework could be measured through a quantitative longitudinal study. Another limitation is the very different ways in which Management 3.0 can be integrated into organizations. Unlike LeSS, which is a framework, and therefore prescribes a set of practices for its implementation, Management 3.0 is a mindset that can be applied in several teams, including and beyond IT. In this sense, and for future work, it becomes important to explore the role of Management 3.0 in various frameworks such as SAFe, Nexus, Spotify, or Scrum@Scale. This could help identify a set of principles advocated in Management 3.0 that might be more easily integrated into a given framework.

**Author Contributions:** Conceptualization, E.E.; methodology, F.A.; validation, E.E.; formal analysis, F.A.; investigation, F.A. and E.E.; writing—original draft preparation, F.A.; writing—review and editing, F.A. and E.E. All authors have read and agreed to the published version of the manuscript.

**Funding:** This research received no external funding.

**Data Availability Statement:** Not applicable.

**Conflicts of Interest:** The authors declare no conflict of interest.

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
