# Peer review of "Adoption of Large-Scale Scrum Practices through the Use of Management 3.0"

_informatics, doi:10.3390/informatics9010020_

Round 1

Reviewer 1 Report

The paper explores how Management 3.0 principles can be applied in the context of the ten practices proposed in the LeSS framework. The content is quite shallow and honestly, I really cannot see the contribution.
If the authors would like to proceed with the work I suggest conducting a deeper analysis of the use cases following the software engineering methodologies.

Author Response

We appreciate the review suggestions and comments received by the reviewer. These elements are key to improving the final quality of the manuscript. Below we respond to each issue raised.

Review #1

The paper explores how Management 3.0 principles can be applied in the context of the ten practices proposed in the LeSS framework. The content is quite shallow and honestly, I really cannot see the contribution.

Author’s response: We thank the reviewer for his/her time in reading this study and for his/her suggestions. We also appreciate the opportunity to clarify the contributions provided by this study. We believe that the contributions made are relevant, but we recognize that it was not adequately presented and explored. Management 3.0 is a practice applied to the area of team management that appeared only in 2011 by proposal of Jurgen Appelo. This is a paradigm that has been applied to young and small companies. However, its adoption in larger companies with large-scale agile teams is a topic that has not been explored in the literature. Therefore, this study becomes relevant to understand the benefits and challenges of adopting Management 3.0 in companies that adopt the Large-Scale Scrum (LeSS) framework. We take this opportunity to review the introduction section and explain the contributions offered by this study.

If the authors would like to proceed with the work I suggest conducting a deeper analysis of the use cases following the software engineering methodologies.

Author’s response: Thank you for your encouragement to proceed with this study. We have reformulated the work and performed significant changes in the manuscript. We essentially highlight three of the most significant changes. First, we have improved the methodological process and presentation of the results. To this end, we have added a Figure 2 that describes how the thematic analysis process was carried out. We have also revised Table 2 to present a quantitative view of the relevance of the themes found for each Less practice. We also made important changes to the discussion section by addressing how Management 3.0 in the context of a LeSS framework can contribute to reducing the main risks associated with migration to large-scale agile. Finally, we reviewed the theoretical and practical contributions of this study to clarify the main contributions and implications of this study. 

Reviewer 2 Report

The authors presented an interesting article to explore how the principles of Management 3.0 can be applied in the context of the ten practices proposed in the LeSS framework. To this end, the authors presented a qualitative research methodology based on four case studies to identify and explore the role of Management 3.0 in management and software development processes that adopt this agile paradigm.

The article is edited correctly.

However, I feel that the authors have not addressed enough the computational part of the ability to present a quantitative analysis. The case study presented in the chapter:
3. Materials and Methods  
... describes the case study in a satisfactory way. However, in my opinion, the authors should propose "their methodology" and "how to solve" the adoption of Scrum practices on a selected example.

Summary

The authors achieved the goal of the article and demonstrated (using numerous literature examples) the role of Management 3.0 in management and software development processes, in particular, the results obtained show that the principles of Management 3.0 are relevant to the implementation of practices within LeSS, especially in terms of supporting team values and personal values.

In addition, different principles between the two paradigms were also identified, namely the greater process rigidity recommended within LeSS and the greater emphasis on process automation.

The authors only presented a QUALITATIVE research methodology based on four case studies to identify and study which adopt this agile paradigm. But in my opinion, a QUANTITATIVE analysis should be done. Using a selected example. I know this can be difficult. But maybe they can compare (in detail) a minimum of two case studies described? 
On the other hand, the article in its current form could be published if it is to be a review article

Author Response

We appreciate the review suggestions and comments received by the reviewer. These elements are key to improving the final quality of the manuscript. Below we respond to each issue raised.

Review #2

The authors presented an interesting article to explore how the principles of Management 3.0 can be applied in the context of the ten practices proposed in the LeSS framework. To this end, the authors presented a qualitative research methodology based on four case studies to identify and explore the role of Management 3.0 in management and software development processes that adopt this agile paradigm.

Author’s response: Thank you very much for your positive feedback and excellent summary of the performed study.

The article is edited correctly. However, I feel that the authors have not addressed enough the computational part of the ability to present a quantitative analysis. The case study presented in the chapter:

  1. Materials and Methods

... describes the case study in a satisfactory way. However, in my opinion, the authors should propose "their methodology" and "how to solve" the adoption of Scrum practices on a selected example.

Author’s response: Thanks for your improvement suggestion. The great advantage of applying a qualitative methodology is that it allows for a comprehensive understanding of the relevance of the Management 3.0 principles applied to the ten practices of the Less framework. However, we recognize that the presentation of the themes identified in each practice provides little detailed information. In this sense, and through the adoption of the webQDA software, we were able to identify the relevance of these practices for each case study. Accordingly, we have restructured the material and methods section to present the thematic analysis methodological process in the Figure 2. Initially, the 72 previously mentioned themes are loaded into webQDA. After that, the software is responsible for finding occurrences in the four case studies. Finally, the number of occurrences is counted, and the themes are sorted in descending order of their incidence level. The degree of homogeneity of each theme found is also analyzed and, for this purpose, its standard deviation is calculated. A high value of the standard deviation indicates high levels of asymmetry among the case studies, while a low value indicates that the theme is typically found homogeneously across all case studies.

The authors achieved the goal of the article and demonstrated (using numerous literature examples) the role of Management 3.0 in management and software development processes, in particular, the results obtained show that the principles of Management 3.0 are relevant to the implementation of practices within LeSS, especially in terms of supporting team values and personal values. In addition, different principles between the two paradigms were also identified, namely the greater process rigidity recommended within LeSS and the greater emphasis on process automation.

Author’s response: Thank you very much for your positive feedback regarding the practical contributions of this work for software companies that implement the LeSS framework.

The authors only presented a QUALITATIVE research methodology based on four case studies to identify and study which adopt this agile paradigm. But in my opinion, a QUANTITATIVE analysis should be done. Using a selected example. I know this can be difficult. But maybe they can compare (in detail) a minimum of two case studies described?

Author’s response: We appreciate the reviewer's input and the suggestion to improve the methodological process. We recognize that all methodologies have vulnerabilities and limitations, but we also agree that it is important to decrease the risk of bias and improve the quality of data and its interpretation.

There is a wide debate in the scientific community about the potentials and limitations of quantitative and qualitative methods. Both can be relevant and robust depending on the methodological rigor of their implementation. Qualitative research is an exploratory methodology. Unlike qualitative research, the focus of quantitative research is on measuring rather than exploring. That is, the goal is to quantify a problem and understand the size of it. Typically, quantitative methodology is mostly employed in areas where the characteristics and dimensions of the research area are well known and limited. The area of Management 3.0 is emerging and its adoption in software companies, especially those that adopt Scrum on a large scale (e.g., LeSS framework) is recent. In this sense, the adoption of the qualitative methodology is more appropriate to understand in depth this phenomenon and the relevance of Management 3.0 principles for each specific practice of the LeSS framework. Nevertheless, and acknowledging the reviewer's contributions, we consider that it would be pertinent to provide more detailed information, of a quantitative nature, on the occurrence of the various themes. Accordingly, we have revised Table 2 to give information about the number of occurrences of each theme and added a new column with information about their degree of distribution for each case study. With this we intend to assess the degree of homogeneity of each theme.

On the other hand, the article in its current form could be published if it is to be a review article

Author’s response: We have made improvements in the methodological process that in our opinion have significantly improved this work, especially in its methodological component. In this sense, we believe that the most appropriate typology for publication is as an article. However, we leave the decision to the editor to consider this work as a review article. We are willing to accept both typologies.

Reviewer 3 Report

The paper shows a study aimed at exploring the potential and importance of using the practices of the Management 3.0 domain in the development of large-scale software projects, in which different development groups participate, while following the principles of agile methodologies. This is, without a doubt, an interesting and potentially useful topic.

The introductory sections (section 1) and related works (section 2) are very well thought out. These sections correctly describe the problem behind the use of agile methodologies in multi-team projects.

However, the paper does not show relevant contributions in the field. The study is based on a qualitative analysis based on interviews (it is not clear if these interviews have been structured) with development managers of 4 companies. An important manuscript defect is the lack of a detailed description of the context and conditions in which software projects are developed in these companies. A description of the structure and approach of the interviews carried out is also missing. In any case, the most important lack is in the conclusions reached. Actually, the discussion and the conclusions do not make a significant contribution to the study of the shortcomings of agile methods in large-scale projects, nor are the lessons learned after the analysis of the interviews included in these sections. To what extent does the study carried out contribute to improving the problem posed? What previously unrecognized problems have come to light?

Author Response

We appreciate the review suggestions and comments received by the reviewer. These elements are key to improving the final quality of the manuscript. Below we respond to each issue raised.

Review #3

The paper shows a study aimed at exploring the potential and importance of using the practices of the Management 3.0 domain in the development of large-scale software projects, in which different development groups participate, while following the principles of agile methodologies. This is, without a doubt, an interesting and potentially useful topic.

Author’s response: Thank you very much for your positive evaluation regarding the interest and usefulness of this study for software community.

The introductory sections (section 1) and related works (section 2) are very well thought out. These sections correctly describe the problem behind the use of agile methodologies in multi-team projects.

Author’s response: Thank you very much again for your very positive feedback regarding the structure of this paper.

However, the paper does not show relevant contributions in the field. The study is based on a qualitative analysis based on interviews (it is not clear if these interviews have been structured) with development managers of 4 companies. An important manuscript defect is the lack of a detailed description of the context and conditions in which software projects are developed in these companies. A description of the structure and approach of the interviews carried out is also missing.

Author’s response: We thank the reviewer's suggestion, and we have reformulated Table 1 with the presentation of the case studies to focus its description on the context and conditions in which the LeSS framework is applied in these companies. Furthermore, we detailed the methodological process, namely the process of structuring and conducting the interviews, and subsequent data analysis. We have included a Figure 2 that describes the methodological process of identifying and analyzing the themes using webQDA software. Moreover, we have reformulated Table 2 to present an accounting of the themes by case study and an analysis of their degree of homogeneity.

In any case, the most important lack is in the conclusions reached. Actually, the discussion and the conclusions do not make a significant contribution to the study of the shortcomings of agile methods in large-scale projects, nor are the lessons learned after the analysis of the interviews included in these sections. To what extent does the study carried out contribute to improving the problem posed? What previously unrecognized problems have come to light?

Author’s response: Thank you very much for your recommendation to clarify and improve the contributions of this study. We have performed two important changes in the manuscript. First, we improve the discussion section by identifying how the adoption of Management 3.0 in the context of organizations adopting the LeSS framework can contribute to mitigating the challenges of large-scale Scrum adoption. Second, we revised the conclusions section to address the main gaps in the literature in this field. We recognize that the topic of agile implementation on a large scale has been little explored since the main studies have mainly focused on the implementation of Agile in small teams that share the same organizational environment. Furthermore, studies concerning the implementation of LeSS have mostly looked at the processes of replicating Scrum from a single team to multiple teams (e.g., at local and distributed levels). Moreover, the mains focus has been the comparative analyzes of the LeSS framework with other alternative models such as SAFe or Nexus are noteworthy. This study takes a different perspective by looking at how practical LeSS can be enhanced by the adoption of Management 3.0. Finally, we have also explored the practical contributions of this work.

We have also included the following references:

Dikert, K.; Paasivaara, M.; Lassenius, C. Challenges and success factors for large-scale agile transformations: A systematic literature review. J Syst Soft 2016, 119, 87-108. https://doi.org/10.1016/j.jss.2016.06.013

Srivastava, P.; Jain, S. A leadership framework for distributed self-organized scrum teams. T Perf Manag 2017, 23, 293-314. https://doi.org/10.1108/TPM-06-2016-0033

Crevani, L.; Uhl-Bien, M.; Clegg, S.; Todnem, R. Changing Leadership in Changing Times II. J Ch Manag 2021, 21, 133-143. https://doi.org/10.1080/14697017.2021.1917489

Reviewer 4 Report

The paper entitled "Adoption of Large-Scale Scrum Practices Through the Use of Management 3.0" explores the principles of Management 3.0 applied in LeSS framework, analysing use cases from real companies. The paper is well structured, focuses on the study of project management methodologies adopted in companies today, and uses an appropriate language style, so it is ready to publish in the present form. Please, check and solve two minor typos before publication:

  • Line 218. A close bracket is missing before the period.
  • Table 1, CS2: The text says that the company was founded in 2014 but it has 9 years of activity (2014+9=2023). 

Author Response

We appreciate the review suggestions and comments received by the reviewer. These elements are key to improving the final quality of the manuscript. Below we respond to each issue raised.

Review #4

The paper entitled "Adoption of Large-Scale Scrum Practices Through the Use of Management 3.0" explores the principles of Management 3.0 applied in LeSS framework, analysing use cases from real companies. The paper is well structured, focuses on the study of project management methodologies adopted in companies today, and uses an appropriate language style, so it is ready to publish in the present form.

Author’s response: Thank you very much for your very positive feedback regarding our study.

Please, check and solve two minor typos before publication:

Line 218. A close bracket is missing before the period.

Table 1, CS2: The text says that the company was founded in 2014 but it has 9 years of activity (2014+9=2023).

Author’s response: Thank you very much for these observations. We have corrected these issues.

- In lines 217-220: In Yin [43] two approaches to implementing multiple case studies are acknowledged, namely the literal replication model (e.g., where the chosen cases are expected to have similar results) and the theoretical replication (e.g., where the chosen cases are expected to have contrasting results, for predetermined reasons).

- Table 1, CS2: The company was founded in 2014 and therefore it has approximately 8 years of activity. We have corrected this information in the manuscript.

Round 2

Reviewer 1 Report

the authors addressed all the issues of the first review round.